# Antitumor and Anti-Invasive Effect of Apigenin on Human Breast Carcinoma through Suppression of IL-6 Expression

**DOI:** 10.3390/ijms20133143

**Published:** 2019-06-27

**Authors:** Hwan Hee Lee, Joohee Jung, Aree Moon, Hyojeung Kang, Hyosun Cho

**Affiliations:** 1College of Pharmacy, Duksung Women’s University, Seoul 132-714, Korea; 2Duksung Innovative Drug Center, Duksung Women’s University, Seoul 132-714, Korea; 3College of Pharmacy, Research Institute of Pharmaceutical Sciences and Institute for Microorganisms, Kyungpook National University, Daegu 702-701, Korea

**Keywords:** apigenin, human breast carcinoma, interleukin-6, antitumor, anti-invasive

## Abstract

Interleukin (IL)-6 plays a crucial role in the progression, invasion, and metastasis of breast cancer. Triple-negative breast cancer (TNBC) cell line MDA-MB-231 is known for its aggressive metastasis. Epithelial to mesenchymal transition (EMT) is a critical process in cancer metastasis. The positive correlation between IL-6 and EMT in tumor microenvironment is reported. We found significantly upregulated IL-6 expression in MDA-MB-231 cells. A blockade of IL-6 expression decreased levels of phosphorylated signal transducer and activator of transcription 3 (pSTAT3), phosphatidylinositol-4,5-bisphosphate 3-kinase (PI3K), phosphorylated protein kinase B (pAkt), and cell cycle-related molecules, including cyclin-dependent kinases (CDKs) and cyclins in MDA-MB-231 cells. A short-hairpin RNA (shRNA)-mediated blockade of IL-6 expression inhibited migration and N-cadherin expression and induced E-cadherin expression in MDA-MB-231 cells. Growth rate was slower for the tumors derived from IL-6 shRNA-treated MDA-MB-231 cells than for those derived from control shRNA-treated MDA-MB-231 cells. The expression of pSTAT3, phosphorylated extracellular signal-regulated kinase (pERK), PI3K, pAkt, snail, vimentin, and N-cadherin was significantly lower in tumors from IL-6 shRNA-treated MDA-MB cells. In addition, apigenin treatment significantly inhibited the growth of MDA-MB-231-derived xenograft tumors along with the protein expressions of pSTAT3, pERK, IL-6, PI3K, pAkt, and N-cadherin. Our results demonstrate that the anti-invasive effect of apigenin in MDA-MB-231-derived xenograft tumors is mediated by the inhibition of IL-6-linked downstream signaling pathway.

## 1. Introduction

Human breast cancers are classified into three subgroups, namely estrogen or progesterone receptor-positive (ER+ or PR+); Her2/neu-positive (HER2/neu+); and triple-negative breast cancer (TNBC) without ER, PR, and Her2/neu expression [1,2]. TNBC accounts for approximately 15% of all breast cancers and is phenotypically characterized with invasiveness, metastasis, and poor prognosis to chemotherapy as compared to other breast cancer types [3]. The past decade has witnessed tremendous therapeutic progress in the targeting of ER+, PR+, or Her2+ breast cancers, but the therapeutic strategies targeting TNBC human breast cancer are still insufficient [3,4]. The MDA-MB-231 cell line lacks the expression of ER, PR, and HER2 [5] and exhibits relatively high aggressiveness and invasiveness. Hence, the MDA-MB-231 cell line has become one of the major cell lines in the study of human breast cancer [5,6]. 

Pro-inflammatory cytokines such as tumor necrosis factor (TNF)-α, interleukin (IL)-1β, and IL-6 are reported to participate in chronic inflammation of tumor microenvironment, thereby creating a favorable environment for tumor progression and metastasis [7]. Of these, IL-6 has been implicated as the key cytokine involved in chronic inflammation in cancer [8,9]. The plasma level of IL-6 is much higher in patients with breast cancer than in healthy donors [10,11] and correlates with frequent metastasis and poor survival in breast cancer [12,13]. ER-negative breast cancers produce more IL-6 than ER-positive breast cancers, and the high expression of IL-6 is linked with higher tumor grades and stages [10,14]. Signal transducer and activator of transcription 3 (STAT3) is selectively activated by IL-6 and thought to be constitutively expressed in more than 50% of breast cancers. The enhanced expression of STAT3 is manifested in ER-negative breast cancers [15] and closely related to the invasiveness and metastasis of cancer cells [16,17]. The overexpression of IL-6 in prostate cancer cell line LNCaP was shown to significantly accelerate the growth of tumor through an increase in the expression of phosphorylated extracellular signal-regulated kinase (pERK) [18]. 

Epithelial to mesenchymal transition (EMT) is a phenotypic change in cell morphology and functional characteristics from epithelial to mesenchymal form [19]. During the process of EMT, epithelial cells reverse morphology into mesenchymal cells and gain increased abilities for migration and invasiveness, which could result in metastasis during cancer progression [19]. The overexpression of IL-6 in ER-positive breast cancer cell lines was shown to induce EMT and to increase the migration and invasion of cells [20]. However, the relationships between a high-level expression of IL-6 and phenotypic changes associated with cell migration or invasion are incompletely investigated in ER-negative breast cancers. 

Apigenin, a flavonoid compound found in plants such as parsley, garlic, and celery, is known to exhibit various biological activities, including antioxidant, anti-inflammatory, antibacterial, antiviral, and anticancer effects [21,22,23,24,25]. The antimetastatic effect of apigenin has been reported in several cancers such as breast, prostate, skin, lung, and ovarian cancers [26,27,28,29,30]. The molecular mechanisms underlying the anticancer or antimetastatic activities of apigenin are thought to involve the targeting of mitogen-activated protein kinase (MAPK)/nuclear factor kappa B, phosphorylated Janus kinase 1 (pJAK1)/STAT3 signaling, vascular endothelial growth factor (VEGF), and aromatase in cancer cells [27,31,32]. The MAPK or JAK/STAT3 signaling pathway is triggered via IL-6 signaling, and VEGF and aromatase are closely related to the IL-6 pathway [32,33,34]. Apigenin was shown to inhibit the production of several inflammatory proteins such as C–C chemokine ligand 2 (CCL2), IL-1α, granulocyte-macrophage colony-stimulating factor (GM-CSF), and IL-6 in MDA-MB-231 cells and may serve as a potential anticancer or antimetastatic agent [35]. In addition, hepatocyte growth factor (HGF)-mediated invasive growth and metastasis of MDA-MB-231 cells was blocked in response to pretreatment of cells with apigenin [36]. 

In the present study, we investigated the role of IL-6 in human breast cancer cell line MDA-MB-231 in terms of tumor growth and metastasis and elucidated the molecular mechanism underlying IL-6-mediated inflammation in cancer metastasis. We also evaluated the effects of apigenin as an anticancer reagent against human breast carcinoma in a xenograft mouse model. 

## 2. Results

### 2.1. The Production of IL-6 is Higher in MDA-MB-231 Cells than in MCF-7 Cells

To compare the level of cytokines released from human breast cancer cell lines MDA-MB-231 and MCF-7, the supernatant from each cell line was harvested after 24 h of cultivation. Secreted cytokines were assessed using Human Cytokine Array C1 as well as an enzyme-linked immunosorbent assay (ELISA). The levels of GM-CSF, IL-6, IL-8, and monocyte chemoattractant protein-1 (MCP-1) were relatively higher in MDA-MB-231 cells than in MCF-7 cells (Figure 1a,b). The expression of IL-6 and IL-8 was detected in both MDA-MB-231 and MCF-7 cells but was higher in MDA-MB-231 cells than in MCF-7 cells. We evaluated the level of IL-6 using ELISA and found significantly higher levels in MDA-MB-231 cells than in MCF-7 cells (Figure 1c). 

### 2.2. Blockade of IL-6 Expression Decreases the Level of pSTAT3, PI3K, and pAkt Proteins in MDA-MB-231 Cells

To investigate the effects of IL-6 expression blockade on the levels of signaling molecules in MDA-MB-231 cells, we treated cells with anti-IL-6 or IL-6 shRNA. The suppression of IL-6 expression using anti-IL-6 antibody in MDA-MB-231 cells decreased the expression of pSTAT3 protein but had minor effects on the expression levels of PI3K, STAT3, ERK, and pERK (Figure 2a). The inhibition of IL-6 expression by IL-6 shRNA resulted in a significant reduction in the expression level of pSTAT3, PI3K, and pAkt, all of which are known to be triggered by IL-6 signaling (Figure 2b). Treatment with IL-6 shRNA also resulted in the change in the cellular morphology to a round form (Figure 2c). 

### 2.3. Blockade of IL-6 Expression Decreases the Levels of CDKs and Cyclins and Induces p21 Expression

IL-6-mediated expression of pSTAT3, PI3K, and pAkt is high during the proliferation of triple-negative breast cancer cells [37,38]. Therefore, we examined the expression of cell proliferation-related molecules in response to blockade of IL-6 expression, such as p53, p21, CDK2, CDK4, CDK1, cyclin D1, and cyclin B1, by a western blot analysis. As shown in Figure 3b, the knockdown of IL-6 expression in MDA-MB-231 cells significantly increased the expression levels of p21 proteins and decreased the expression levels of CDKs (CDK2, CDK4, and CDK1) and cyclins (cyclin D1 and cyclin B1) (Figure 3a,b). 

### 2.4. Blockade of IL-6 Expression Inhibits Cell Invasion and Metastasis Factors in MDA-MB-231 Cells 

To investigate the anti-invasive effect in response to the blockade of IL-6 expression in MDA-MB-231 cells, we evaluated the invasiveness of cells and expression of EMT-related molecules such as E-cadherin and N-cadherin. As shown in Figure 4a, the invasiveness of MDA-MB-231 cells decreased in response to treatment with anti-IL-6 or IL-6 shRNA. Furthermore, the expression of E-cadherin increased and that of N-cadherin significantly decreased in MDA-MB-231 cells transfected with IL-6 shRNA (Figure 4b). A meta-analysis-based biomarker assessment with Kaplan–Meier Plotter [39] revealed the negative correlation between the blood level of IL-6 and relapse-free survival in patients with TNBC. IL-6 level showed a negative association with relapse-free survival of patients with lymph node metastasis or grade 3 tumors (Figure 4c).

### 2.5. Blockade of IL-6 Expression Inhibits the Growth of MDA-MB-231-Derived Tumors 

BALB/c nude mice were subcutaneously implanted with 5 × 10^6^ MDA-MB-231 cells transfected with control (ctrl) shRNA or IL-6 shRNA. After 14 days, tumor was measured every other day until the volume reached about 1000–1200 mm^3^. Figure 5a,b shows that IL-6 shRNA treatment of MDA-MB-231 cells significantly inhibited the tumor growth as compared with control shRNA treatment. We examined the expression levels of molecules related to IL-6 signaling in the tumor tissues derived from MDA-MB-231 cells. The JAK/STAT3, MAPK, and PI3K-Akt-mTOR pathways are triggered by IL-6 signaling and are associated with tumor cell growth, proliferation, and metastasis [40]. Recent reports have shown that the knockdown of pSTAT3 expression resulted in the reduction in the growth and tumor formation ability of TNBC cells [37]. As seen in Figure 5c,d, the expression of pSTAT3, pERK, PI3K, and pAkt proteins was significantly lower in the tumor tissues derived from IL-6 shRNA-treated MDA-MB-231 cells than in those derived from control shRNA-treated MDA-MB-231 cells. Furthermore, a considerable decrease in the expression levels of snail, vimentin, and N-cadherin was observed in the tumor tissues derived from IL-6 shRNA-treated MDA-MB-231 cells. The reduced expression levels of pSTAT3, pAkt, and N-cadherin proteins were also confirmed by immunohistochemistry (IHC) staining of tumor tissues derived from the xenograft mice implanted with IL-6 shRNA-treated MDA-MB-231 cells (Figure 5d). 

### 2.6. Treatment of MDA-MB-231 Cells with Apigenin Decreases the Expression Level of Snail and N-cadherin via IL-6 Inhibition

Apigenin is a flavonoid compound with anticancer effects on breast cancer cells that are mediated via apoptotic induction and cell cycle regulation [41,42]. However, studies reporting the antimetastatic effect of apigenin are few, and the molecular mechanism underlying the inhibitory effect of apigenin on cancer cell metastasis is incompletely understood. We investigated whether apigenin (Figure 6a) inhibits the production IL-6 and exerts anti-invasive effects via the suppression of IL-6 expression in MDA-MB-231 cells. Figure 6a,b shows that IL-6 production from MDA-MB-231 cells was significantly inhibited following apigenin treatment in a dose-dependent manner (Figure 6b,c). The expression levels of snail and N-cadherin in MDA-MB-231 cells decreased after apigenin treatment (Figure 6d). The migration and invasiveness of MDA-MB-231 cells clearly decreased in response to apigenin treatment in a dose-dependent manner (Figure 6e,f). In particular, treatment with 20 and 40 μM apigenin resulted in a significant decrease in the migration and invasiveness of MDA-MB-231 cells. 

### 2.7. The Inhibitory Effect of Apigenin on MDA-MB-231-Derived Tumor Growth and Invasiveness is Mediated through Reduced Expression of pSTAT3, pERK, IL-6, and pAkt

To evaluate the antitumor and anti-invasive effects of apigenin in vivo, BALB/c nude mice were subcutaneously injected with 5 × 10^6^ MDA-MB-231 cells. After 14 days, mice were administrated drinking water or 25 or 50 mg/kg of apigenin for another 14 days. The growth of tumor was measured every other day until the tumor volume reached about 1000–1200 mm^3^. The growth rate of tumor was significantly inhibited in mice treated with both the doses of apigenin (Figure 7a,b). Apigenin treatment significantly decreased the expression levels of pSTAT3, pERK, PI3K, and pAkt proteins in the tumor tissues derived from MDA-MB-231-implanted xenograft models (Figure 7c,d). We confirmed the decrease in the expression levels of pSTAT3, pAkt, and N-cadherin by IHC staining of the tumor tissues derived from MDA-MB-231-implanted xenograft mice that were orally administrated with apigenin (Figure 7e). 

## 3. Discussion

The present study demonstrates the inhibitory effects of apigenin through the suppression of IL-6 expression on tumor progression and invasiveness of MDA-MB-231 cells in vitro and in vivo. The overexpression of IL-6 in MDA-MB-231 cells has been reported in many studies [20,43,44], but the implications of IL-6 expression in breast cancer are yet unclear. MCF-7 is a noninvasive breast cancer cell line, whereas MDA-MB-231 is known as a highly invasive breast cancer cell line [5]. We confirmed the higher expression of IL-6 in MDA-MB-231 cells than in MCF-7 cells, suggestive of the positive correlation between the expression of IL-6 and invasiveness of MDA-MB-231 cells (Figure 1a,b). Many studies have reported that chronic inflammation is closely related to tumor progression, migration, and metastasis in various cancers [45,46,47]. Cancer inflammatory environment may be established by several factors, including pro-inflammatory cytokines such as TNF-α, IL-1β, and IL-6 [7]. Of these, IL-6 released from MDA-MB-231 cells was reported to be associated with chronic inflammation [8,48] and is known to selectively induce the activation of the JAK/STAT3 and PI3K/Akt pathways [49]. IL-6-linked JAK/STAT3 and PI3K/Akt signaling pathways have been implicated in cancer progression [50,51,52]. We investigated the effect of knockdown of IL-6 expression using anti-IL-6 or IL-6 shRNA on the expression of inflammation-linked signaling molecules in MDA-MB-231 cells. As a result, we found that the neutralization of IL-6 expression or IL-6 shRNA treatment significantly inhibited the expression level of pSTAT3 (Figure 2a). However, the expression of PI3K and pAkt was downregulated only after treatment with IL-6 shRNA (Figure 2b). Neutralization of IL-6 using anti-IL-6 was probably insufficient to completely block IL-6 secretion, as low levels of IL-6 may be continuously released from MDA-MB-231 cells. 

The interaction between IL-6 and IL-6R was shown to induce antiapoptotic effects on human esophageal carcinoma and multiple myeloma through the phosphorylation of gp130 and STAT3 [53,54]. The JAK/STAT3, MAPK, and PI3K/Akt/mTOR pathways are activated by IL-6 signaling and are closely related to cell growth and metastasis in TNBC [55]. In particular, STAT3, a transcription factor stimulated by IL-6, was shown to be consistently expressed in most breast cancer cells. The expression of the phosphorylated form of STAT3 is high in TNBC and thought to be related to the aggressiveness and invasiveness of tumors [15,56]. Blocking the expression of STAT3 by shRNA treatment in mouse breast cancer cells was shown to inhibit tumor growth, progression, and metastasis [37]. In addition, the activation of the PI3K-pAkt-mTOR pathway triggered by IL-6 was shown to be related to cell proliferation and metastasis in a variety of cancers [57,58]. In our study, we investigated the effect of IL-6 expression blockade by shRNA treatment on the expression of the molecules related to cell cycle progression in MDA-MB-231 cells. As shown in Figure 3a,b, the downregulation of IL-6 expression significantly increased the expression level of p21 (Figure 3a,b). Steiner demonstrated that the overexpression of IL-6 in prostate cancer resulted in a significant increase in cell proliferation through the upregulated expression of CDK2 and increased expression of pERK [18]. The results of our study suggest similar phenotypic characteristics in TNBC with respect to IL-6-mediated expression of pSTAT3, pERK, and cell cycle-associated molecules (Figure 2 and Figure 3). 

Studies have highlighted the crucial role of EMT in cancer metastasis [19]. EMT is known to correlate with increased cell migration and invasiveness [59]. Cancer cells undergoing the EMT process exhibit upregulated expression of N-cadherin, vimentin, fibronectin, and snail and downregulated expression of E-cadherin [60]. The circulating high levels of IL-6 in patients with breast cancer were shown to be positively associated with cancer metastasis and poor survival rate [10,11], consistent with the results of meta-analysis from a Kaplan–Meier Plotter (Figure 4c). We examined whether blocking IL-6 expression may inhibit the growth and metastasis of breast cancer in vivo using a xenograft mouse model. Blockade of IL-6 expression significantly delayed the growth of MDA-MB-231-derived tumor (Figure 5a,b). The tumor tissues derived from IL-6 shRNA-treated MDA-MB-231 cells showed decreased expression of pSTAT3, pERK, PI3K, and pAkt proteins (Figure 5c). The expression levels of snail, vimentin, and N-cadherin proteins, representative biomarkers of EMT process, were also significantly inhibited in the tumors derived from IL-6 shRNA-treated MDA-MB-231 cells (Figure 5c), indicating that IL-6 expression is directly associated with tumor metastasis through the regulation of snail, vimentin, and N-cadherin expression. 

Apigenin is known to have many biological activities, including anti-inflammatory, antioxidant, and anticancer effects [21,22,61]. The anticancer effects of apigenin have been reported in several cancer types, including breast cancer [41,61,62]. Anticancer activities are generally mediated via antiproliferative, antiapoptotic, and/or antimetastatic effects on cancer cells. Very few studies have reported the antimetastatic activity of apigenin [63,64]. Furthermore, the molecular mechanism underlying the antimetastatic effects of apigenin is still under investigation. The expression of TNF-α, IL-1α, and IL-6 from MDA-MB-231 cells was shown to be inhibited by apigenin [35], and the inhibition of TNF-α-induced CCL2 release by apigenin was thought to suppress tumor migration and metastasis through the regulation of tumor microenvironment [35]. We found that apigenin significantly inhibited the production of IL-6 from MDA-MB-231 cells in a dose-dependent manner (Figure 6b,c). We also observed the inhibitory effect of apigenin on the migration and invasion of MDA-MB-231 cells (Figure 6e,f) and reported, for the first time, the significant delay in the growth of tumors in MDA-MB-231 xenograft mice orally treated with apigenin (Figure 7a,b). This effect was directed through the inhibition of IL-6-mediated pSTAT3, pERK, PI3K, and pAkt expression as well as EMT signal-linked N-cadherin expression (Figure 7c–e). The inhibitory effect of apigenin on hepatocyte growth factor-promoted metastasis of MDA-MB-231 cells was shown to be involved in the suppression of the PI3K/Akt pathway; this observation is partly in agreement with the results of the present study [36]. 

In summary, we demonstrate that the blockade of IL-6-associated inflammation positively correlates with the inhibition of tumor growth and EMT process in human breast cell line MDA-MB-231 and that oral administration of apigenin results in the suppression of IL-6-related downstream signaling pathways via anticancer and anti-invasive effects. 

## 4. Materials and Methods 

### 4.1. Specimen Preparation

Apigenin (≥95% pure) was purchased from Sigma-Aldrich, Inc. (St. Louis, MO, USA) and dissolved in dimethyl sulfoxide (DMSO; Sigma-Aldrich, St. Louis, MO, USA), followed by dilution in a medium or drinking water for in vitro or in vivo experiments, respectively. 

### 4.2. Cell Lines and Treatments

Human breast cancer cell lines MDA-MB-231 and MCF-7 were obtained from American Type Culture Collection (ATCC; Manassas, VA, USA). MDA-MB-231 and MCF-7 cells were cultured in Roswell Park Memorial Institute (RPMI)-1640 medium (Corning Inc, New York, NY, USA) and Eagle’s minimum essential medium (EMEM, ATCC, Manassas, VA, USA), respectively, supplemented with 10% heat-inactivated fetal bovine serum (FBS, Young-In Frontier, Seoul, South Korea) and 100 U/mL of penicillin and streptomycin (Gibco, Grand Island, NY, USA) at 37 °C in a humidified atmosphere of 5% CO_2_. Cells were cultured at a density of 5 × 10^5^ cells in 6-well plate overnight and treated with three different concentrations of apigenin (0 μM, 50 μM, and 100 μM) for 6 h.

### 4.3. Human Cytokine Array 

Cytokines released from cells were analyzed using Human Cytokine Array C1 (RayBiotech, Norcross, GA, USA). All experiments were performed according to manufacturer’s instructions. Briefly, membranes were treated with a blocking buffer for 30 min and incubated overnight in the supernatants harvested from cells at 4 °C on a rocker. The membranes were washed with a wash buffer and incubated in biotinylated antibody cocktail diluted in blocking buffer for 2 h at room temperature. Following incubation, the membranes were washed and treated with horseradish peroxidase (HRP)-streptavidin for 2 h at room temperature. The membranes were washed and incubated in a detection buffer for 2 min at room temperature. The spots were analyzed using ChemiDoc (Protein Simple, San Jose, CA, USA). 

### 4.4. Real-Time Quantitative Polymerase Chain Reaction (RT-qPCR)

The level of mRNA was analyzed by RT-qPCR using SYBR Green (BioLine, London, UK). Total cellular RNA was extracted with TaKaRa MiniBEST Universal RNA Extraction Kit (#9767, TaKaRa, Kusatsu, Japan), and cDNA synthesis was conducted using PrimeScript^TM^ 1st Strand cDNA Synthesis Kit (#6110A, TaKaRa, Shiga, Japan) in accordance with the manufacturer’s instructions. Briefly, cells were treated as indicated and harvested for total RNA extraction. The extracted RNA was quantitated with Nanodrop (MicroDigital Co., Seoul, South Korea), and 1 μg of RNA was used for 50-ng/μL cDNA synthesis. RT-qPCR was performed under the following conditions: 20 s at 94 °C for denaturation, 20 s at 60 °C for annealing, and 1 min at 72 °C for extension for 40 cycles. PCR primer sequences used were as follows: human IL-6F (5′-GCC CAG CTA TGA ACT CCT TCT-3′) and IL-6R (5′-GAA GGC AGC AGG CAA CAC-3′).

### 4.5. Short-Hairpin RNA (shRNA) Lentivirus Plasmid Transfection Using Lipofectamine 2000 

Human IL-6 shRNA lentivirus plasmid with pGFP vector kit (#TL312162, Origene, Rockville, MD, USA) was used for blocking IL-6 expression in MDA-MB-231 cells. Briefly, cells were seeded at a density of 5 × 10^5^ cells/well in a six-well plate and incubated overnight. After incubation, the supernatant was aspirated, and the cells were treated with the mixture of shRNA plasmid-Lipofectamine 2000 reagent (Invitrogen, Carlsbad, CA, USA) for 5 h at 37 °C. After incubation, the medium was replaced with fresh medium supplemented with 10% FBS and 100 U/mL of penicillin-streptomycin, and the cells were cultured at 37 °C. For selection, cells were incubated in a medium containing puromycin at appropriate concentration.

### 4.6. Invasion Assay

Invasive capability of MDA-MB-231 cells was evaluated with a transwell system (Corning Inc, New York, NY, USA). Briefly, a transwell insert was coated with 2 mg/mL of Matrigel (Corning Inc, New York, NY, USA). Cells were seeded into the upper chamber of the transwell in a 200-μL volume of medium with serum at a density of 2–5 × 10^5^ cells/mL. The lower chamber was filled with medium containing 10% serum. The cells were incubated for 12 or 24 h at 37 °C. After incubation, the medium was aspirated and the cells were washed twice with Dulbecco’s phosphate-buffered saline (DPBS). The cells were fixed with 100% methanol for 30 min at room temperature, followed by washing and staining with 0.2% crystal violet (Sigma-Aldrich, St. Louis, MO, USA) diluted in DPBS for 30 min at room temperature in the dark. The invasive cells were counted under a light microscope (×100). Apigenin was added into the lower chamber of the transwell. 

### 4.7. Scratch Motility Assay

Briefly, cells were plated at a density of 2 × 10^5^ cells/well in a 24-well plate and incubated overnight. A wound was created by making a scratch (1 mm width). The cells were washed twice with DPBS and incubated with fresh medium. Cells were treated with several concentrations (0, 10, 20, and 40 μM) of apigenin for different time points (0, 6, 12, and 24 h). Images were captured under a light microscope (×100). 

### 4.8. Western Blot Analysis 

Cells were seeded at a density of 5 × 10^5^ cells/well in a six-well plate and treated with various concentrations of apigenin (0, 10, 20, and 40 μM) for 24 h. Cells were lysed using protein extraction buffer (Intron Biotechnology, Seoul, South Korea) supplemented with phosphatase inhibitor (genDEPOT, Katy, TX, USA). Proteins extracted from cells were quantified with Coomassie (Bradford) Protein Assay (genDEPOT, Katy, TX, USA), separated by electrophoresis, and transferred onto nitrocellulose membranes (0.45-μM pore size, Merck Millipore, Burlington, MA, USA). The membranes were blotted with first and second antibodies. PI3K (#4255), pAkt (#4060), STAT3 (#4904) (and phopho-STAT3, Ser727 and Tyr701, #9134 and #9167), ERK (#9102) (and phopho-ERK, #4370), snail (#3870), vimentin (#5741), and N-cadherin (#13116) antibodies, and a cell cycle regulation sampler kit (#9932) was obtained from Cell Signaling Technology (CST, Danvers, MA, USA). Anti-β-actin (#A5441, Sigma Aldrich, St. Louis, MO, USA), anti-p53 (#05-224, Millipore, Burlington, MA, USA), and anti-IL-6 (Abcam, Cambridge, UK) were used. The second antibodies (anti-mouse and anti-rabbit) were purchased from Santa Cruz Biotechnology (Dallas, TX, USA). The blots were visualized with an enhanced chemiluminescent (ECL) detection solution (Intron Biotechnology, Seoul, South Korea) 

### 4.9. In Vivo Experiment Using Xenograft BALB/c Nude Mice 

All animal experiments were conducted in accordance with National Research Council’s Guide (IACUC, Seoul, Republic of Korea) for the Care and Use of Laboratory Animals. The experimental protocol was approved by the Animal Experiments Committee of Duksung Women’s University (permit number: 2018003007). BALB/c nude mice (female, 5-week old) were obtained from Raon Bio Co. Ltd. (Seoul, South Korea). Mice were acclimatized in specific pathogen free (SPF) environment and had access to food and water under a 12-h day/12-h night period at 23–27 °C. Mice were grouped and subcutaneously implanted with 5 × 10^6^ MDA-MB-231 cells or shRNA-transfected MDA-MB-231 cells into the dorsum next to the right hind leg. After 14 days, the tumors of mice were measured every other day using standard caliper; tumor size was calculated using the formula (tumor length (mm) × width (mm)^2^)/2, as previously described [65,66]. To evaluate the antitumor effects of apigenin, another set of mouse group were prepared and they were subcutaneously implanted with 5 × 10^6^ MDA-MB-231 cells. After 14 days, mice were orally administrated with drinking water or two doses of apigenin (25 or 50 mg/kg) for another 2 weeks. 

### 4.10. Immunohistochemistry (IHC) Staining of Tumor Tissues

Tumor tissues excised from mice were frozen with Frozen Section Compound (Leica, Wetzlar, Germany) and stored at −80 °C. Tumor tissues were sliced at 0.5-mm widths and placed on slides. The tissue sections were hydrated in 70% ethanol for 5 min and permeabilized in 3% hydrogen peroxide (H_2_O_2_) diluted in methanol for 10 min. Tissues were washed in tap water for 10 min, followed by three washes in 1× PBS for 5 min and incubation in a blocking buffer (10% bovine serum albumin (BSA) + 0.05% Tween-20 in PBS) at room temperature. After 1 h, the tissues were blotted with the first antibody in PBST overnight at 4 °C, followed by the treatment with the second and third antibodies (;ABC kit, VECTASTAIN, Burlingame, CA, USA) for 2 h at room temperature. The blotted tissues were washed thrice in 1× Tris-buffered saline (TBS) for 5 min. Tissues were then stained with DAB (Vector Laboratories, Burlingame, CA, USA) and visualized under a microscope (×200). The first antibodies, pAkt (#4060), pSTAT3 (#9134), and N-cadherin (#13116), were purchased from Cell signaling technology (CST, Danvers, MA, USA), while the second antibody used was mouse anti-rabbit IgG-HRP (#sc-2537, Santa Cruz, Dallas, TX, USA). 

### 4.11. Statistical Analysis

Data were processed using Microsoft Excel, and the results are presented as mean ± standard deviation (SD). Comparisons of several means were performed with one-way analysis of variance followed by Fisher’s Least Significant Difference as a post hoc test. Differences among groups were considered significant at a value of *p* < 0.05.

## Figures and Tables

**Figure 1 ijms-20-03143-f001:**
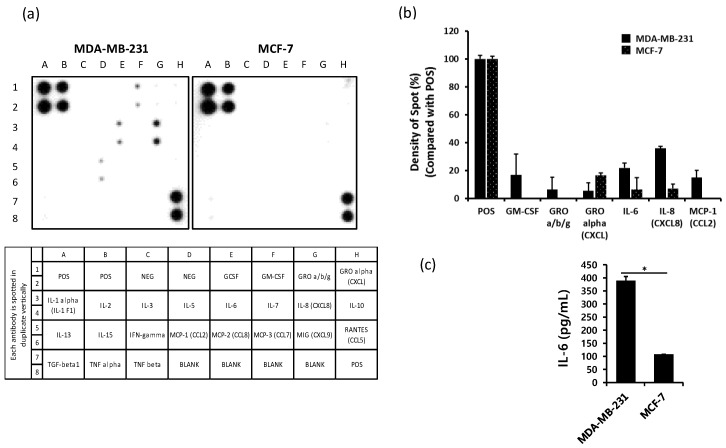
The production of IL-6 is higher in human breast cancer cell line MDA-MB-231 than in MCF-7. MDA-MB-231 and MCF-7 cells were seeded in six-well plates at a density of 5 × 10^5^ cells/well and incubated overnight. After incubation, the supernatants were harvested for Cytokine array assay or ELISA. (**a**) Production of cytokines from MDA-MB-231 and MCF-7 cells. (**b**) Density of spots as compared with positive control (POS). (**c**) Production of IL-6 from MDA-MB-231 and MCF-7 cells. Significant difference is shown: * *p* < 0.05. NEG, negative control; GCSF, granulocyte colony-stimulating factor; GM-CSF, granulocyte-macrophage colony-stimulating factor; GRO a/b/g, growth-regulated oncogene-a/b/g.

**Figure 2 ijms-20-03143-f002:**
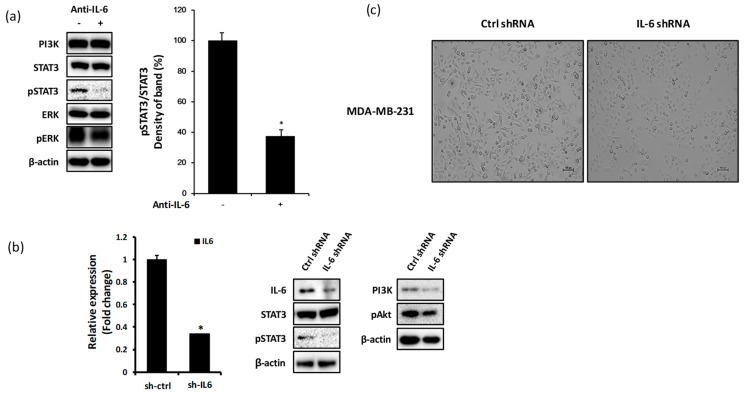
Blockade of IL-6 expression decreases the levels of pSTAT3, PI3K, and pAkt in MDA-MB-231 cells. MDA-MB-231 cells were treated with anti-IL-6 antibody or IL-6 shRNA. After 24 h of incubation, cell lysates were harvested and analyzed by western blotting. (**a**) Expression of PI3K, STAT3 (and pSTAT3), and ERK (and pERK) proteins. (**b**) Expression of IL-6, STAT3 (and pSTAT3), PI3K, and pAkt proteins. (**c**) Morphology of MDA-MB-231 cells (scale value is 100px). Significant difference is shown: * *p* < 0.05.

**Figure 3 ijms-20-03143-f003:**
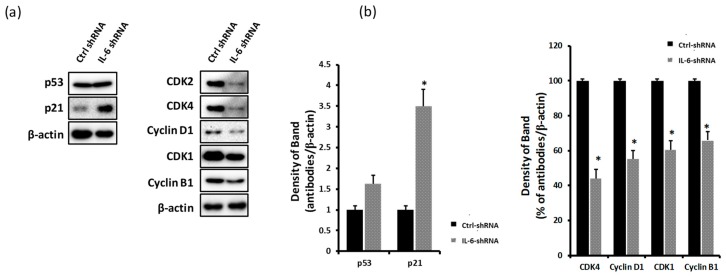
Blockage of IL-6 expression decreases cyclin-dependent kinases (CDK) and cyclin protein expression and induces p21 expression in MDA-MB-231 cells. MDA-MB-231 cells were seeded in six-well plates at a density of 5 × 10^5^ cells/well and treated with IL-6 shRNA or control shRNA. Cell lysates were harvested and analyzed by western blotting. (**a**) Expression of p53 and p21 proteins. (**b**) Expression of CDKs (CDK2, CDK4, and CDK1) and cyclins (cyclin D1 and cyclin B1). Significant difference is shown: * *p* < 0.05.

**Figure 4 ijms-20-03143-f004:**
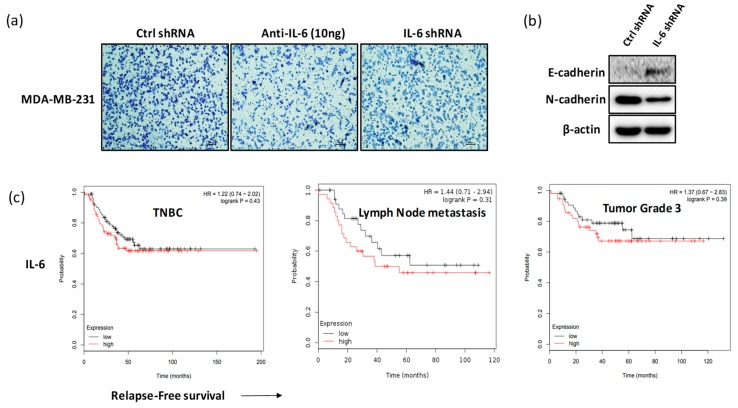
Blockade of IL-6 expression inhibits invasion and metastasis factors in MDA-MB-231 cells. MDA-MB-231 cells were seeded at a density of 2 × 10^5^ cells/mL in a transwell and treated with anti-IL-6 antibody or IL-6 shRNA. After 12 h of incubation, cells were stained with crystal violet dye for invasion assay and cell lysates were harvested for western blotting. (**a**) Invasive capability of MDA-MB-231 cells (scale value is 100px). (**b**) Expression of E-cadherin and N-cadherin proteins. (**c**) Meta-analysis-based biomarker assessment with Kaplan–Meier Plotter showed a negative correlation between the blood level of IL-6 and relapse-free survival in patients with Triple-negative breast cancer (TNBC).

**Figure 5 ijms-20-03143-f005:**
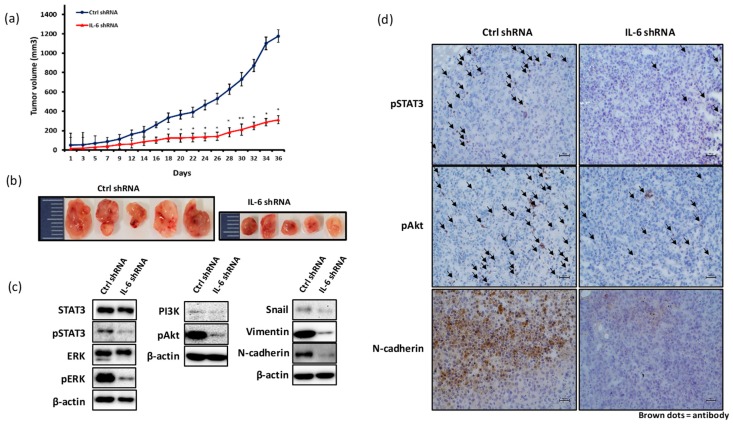
Blockade of IL-6 expression inhibits MDA-MB-231-derived tumor growth and metastasis. MDA-MB-231 cells were transfected with IL-6 shRNA or control shRNA lentivirus plasmid. BALB/c nude mice were subcutaneously injected with 5 × 10^6^ MDA-MB-231 cells transfected with IL-6 shRNA or control shRNA into the dorsum next to right hind leg. After 14 days, tumor volume was measured using caliper every other day. (**a**) Tumor growth curve. (**b**) Tumor photograph. Expression of proteins in tumor tissues analyzed with western blotting and immunohistochemistry (IHC). Expression of STAT3 (and pSTAT3), ERK (pERK), PI3K, pAkt, snail, vimentin, and N-cadherin (**c**). Black arrows indicate the dots of pSTAT3, pAkt, and N-cadherin (**d**) (scale value is 100px). Significant differences are shown: * *p* < 0.05, ** *p* < 0.001.

**Figure 6 ijms-20-03143-f006:**
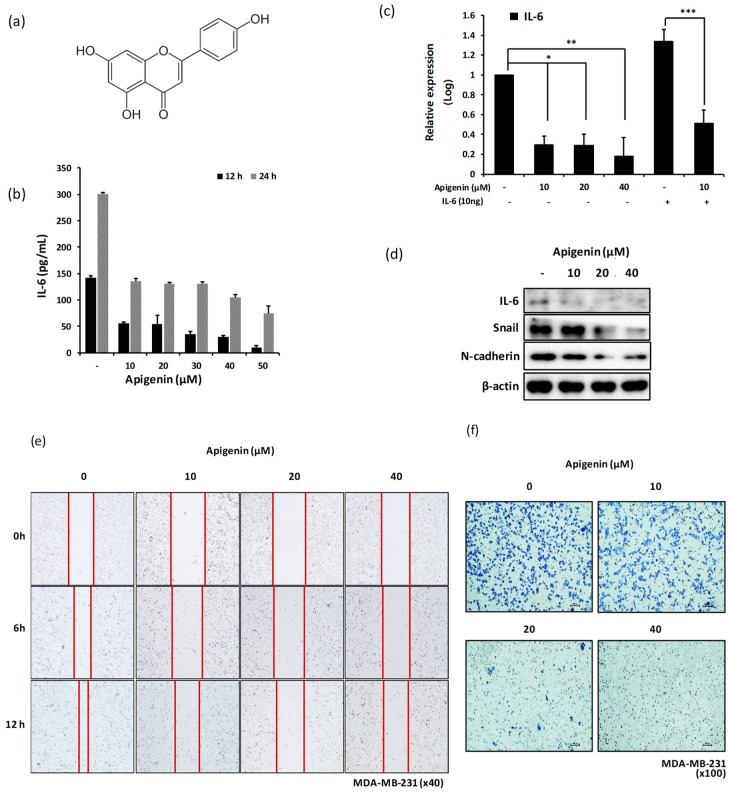
Apigenin treatment decreases the expression of snail and N-cadherin via IL-6 inhibition in MDA-MB-231 cells. MDA-MB-231 cells were plated at a density of 5 × 10^5^ cells/well in a six-well plate and incubated with different concentrations of apigenin (10 to 50 μM) for 24 h. The cells were harvested for western blot analysis and qPCR, and the supernatants were used for ELISA. (**a**) The structure of apigenin. (**b**) Production of IL-6 by ELISA. (**c**) Relative mRNA expression of IL-6 by qPCR. (**d**) Expression of IL-6, snail, and N-cadherin proteins by western blot analysis. Migration and invasiveness of MDA-MB-231 cells were assessed using scratch motility assay and invasion assay, respectively. (**e**) Migratory capability of MDA-MB-231 cells. For scratch motility assay, cells were plated at a density of 2 × 10^5^ cells/well in a 24-well plate and treated with different concentrations (0, 10, 20, and 40 μM) of apigenin for different time points (0, 6, and 12 h). (**f**) Invasive capability of MDA-MB-231 cells. For invasion assay, a transwell insert was coated with Matrigel. Cells were seeded in the upper chamber of transwell at a density of 2–5 × 10^5^ cells/mL and treated with different concentrations of apigenin (0, 10, 20, and 40 μM). After 24 h of incubation, cells were stained with crystal violet (scale value is 100px). Significant differences are shown: * *p* < 0.05, ** *p* < 0.001, *** *p* < 0.002.

**Figure 7 ijms-20-03143-f007:**
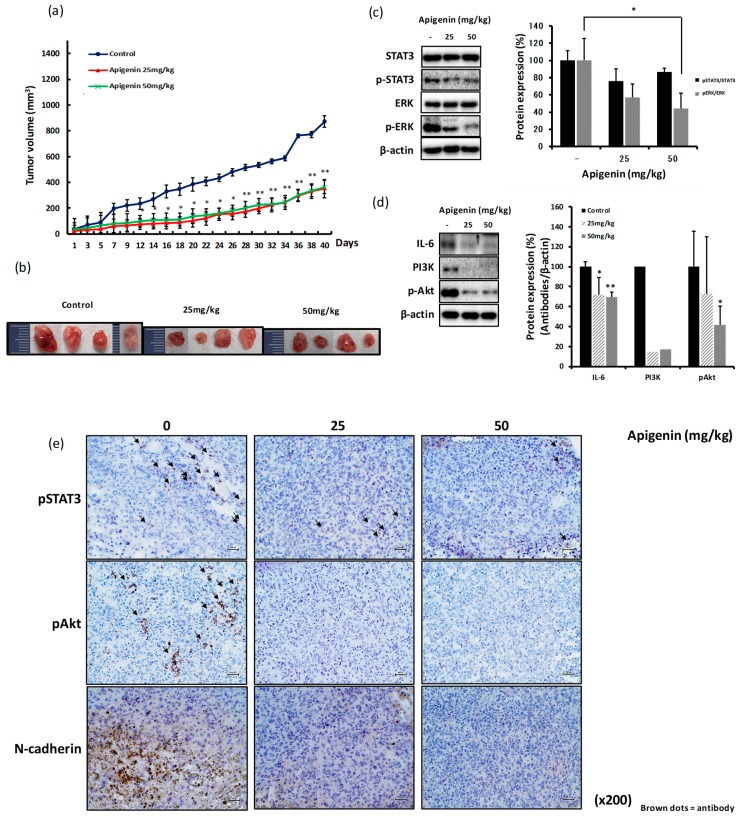
The inhibitory effect of apigenin on the growth and metastasis of a MDA-MB-231 cell-derived tumor is mediated through the reduced expression of pSTAT3, pERK, IL-6, and pAkt. BALB/c nude mice were subcutaneously injected with 5 × 10^6^ MDA-MB-231 cells into the dorsum next to right hind leg. After 14 days, the mice were orally administrated drinking water or apigenin (25 or 50 mg/kg) every day for another 2 weeks. Tumor size was measured by caliper every other day. (**a**) Tumor growth curve. (**b**) Tumor photograph. (**c,d**) Expression of proteins in tumor tissues analyzed by western blotting. (**e**) Expression of proteins in tumor tissues analyzed by immunohistochemistry (IHC). Expression of STAT3 (and pSTAT3), ERK (pERK), PI3K, and pAkt proteins (c,d). Black arrows indicate the dots of pSTAT3, pAkt, and N-cadherin (e) (scale value is 100px). Significant differences are shown: * *p* < 0.05, ** *p* < 0.001.

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
