# Peer review of "Antitumor and Anti-Invasive Effect of Apigenin on Human Breast Carcinoma through Suppression of IL-6 Expression"

_ijms, 2019, doi:10.3390/ijms20133143_

Round 1

Reviewer 1 Report

This isa very interesting study which shows the mechanism of IL-6 depended on the aggression of breast tumour. I do not know why authors decided to include the apigenin study in this paper I suggest add additional studies a separate this to other paper. Other issues:

Language (style and grammar) has very low value, it needs to be verified by a native proofreader

The last sentence in the abstract is not clear

In abstract is lack of conclusions

Line 43 should be TNBC human breast cancer

In introduction when authors present the aim of the study, a description of obtained results should be not given

Figure 4c should  have a better quality

Number of bioethics commission must be provided

In methods should be a detailed description of apigenin treatment

Why Authors decided use Fisher test?

Author Response

This is a very interesting study which shows the mechanism of IL-6 depended on the aggression of breast tumor. I do not know why authors decided to include the apigenin study in this paper I suggest add additional studies a separate this to other paper.

Thank you for the valuable comment, we agree partially with your opinion about excluding apigenin study in this manuscript. However, the special issues that this manuscript was submitted is called “Perspectives on the Health Benefits of Flavonoids”. Therefore, we thought that apigenin study as well as the anti-invasive mechanism of apigenin (through suppression of IL-6) would be well matched. However, we are willing to take out the apigenin study if reviewer insists on separating the data of apigenin.

Other issues:

Language (style and grammar) has very low value, it needs to be verified by a native proofreader

Thank you for the comment, in fact, we attached a certificate of English editing for the manuscript during our initial submission. If the reviewer suggests more editing process, we are happy to carry out another round of editing. 

The last sentence in the abstract is not clear. In abstract is lack of conclusions.

We corrected the last sentence in abstract and added another sentence for conclusions.

 In addition, apigenin treatment significantly inhibited the growth of MDA-MB-231-derived xenograft tumors along with the protein expressions of pSTAT3, pERK, IL-6, PI3K, pAkt, and N-cadherin. Our results demonstrate that anti-invasive effect of apigenin in MDA-MB-231-derived xenograft tumors is mediated by inhibition of IL-6 –linked downstream signaling pathway.

Line 43 should be TNBC human breast cancer

We corrected as you suggested.

In introduction when authors present the aim of the study, a description of obtained results should be not given.

A description of obtained results was removed from introduction as you suggested.

Figure 4c should have a better quality

Unfortunately, figure 4c was automatically generated in online program of KM Meier plotter. Therefore, we can not obtain a better image. If the reviewer think that the quality of image is too poor to be published, we are willing to take out figure 4c from our manuscript.

Number of bioethics commission must be provided

In material and methods section for “In vivo experiment using xenograft BALB/c nude mice”, number of bioethics commission is provided as following.

All animal experiments were conducted in accordance with National Research Council’s Guide (IACUC, Republic of Korea) for the Care and Use of Laboratory Animals. The experimental protocol was approved by the Animal Experiments Committee of Duksung Women’s University (permit number: 2018003007).

In methods should be a detailed description of apigenin treatment

In material and methods section for “cell lines and treatments” and “In vivo experiment using xenograft BALB/c nude mice”, the following descriptions are added individually.

Cells were cultured at a density of 5x105 cells in 6-well plate overnight and treated with three different concentrations of apigenin (0 μM, 50 μM, 100 μM) for 6 hours.

To evaluate the anti-tumor effects of apigenin, another set of mouse group were prepared and they were subcutaneously implanted with 5 ´ 106 MDA-MB-231 cells. After 14 days, mice were orally administrated with drinking water or two doses of apigenin (25 or 50 mg/kg) for another 2 weeks.

Why Authors decided use Fisher test?

We are very sorry for misspelling. We meant Fisher’s Least Significant Difference (LSD), which was used as one of post hoc tests. Statistical analysis part is now revised as following.

Data were processed using Microsoft Excel and results are presented as mean ± standard deviation (SD). Comparisons of several means were performed with one-way analysis of variance followed by Fisher’s Least Significant Difference as a post hoc test. Differences among groups were considered significant at a value of p < 0.05.

Reviewer 2 Report

In their manuscript, the authors described their studies that MDA-MB-231 cells express high level of IL-6, which is critical for cell growth and invasion. They confirmed their results in mice, and found effects of apigenin on human breast carcinoma may be related to suppression of IL-6 expression. The study is interesting and provided large amount of data.

Questions:

The authors observed several cytokines were increased in MDA-MB-231 cells, why did they only further studies IL-6? The level of IL-8 is much higher than IL-6.

Fig.5b and Fig.7b. Tumor photographs were not taken at the same time, and only one image in Fig.5b showed scale. In Fig.7b, tumors of the same groups seem from several different photographs. The authors should show all individual scales if tumors were not in the same photograph.

Author Response

In their manuscript, the authors described their studies that MDA-MB-231 cells express high level of IL-6, which is critical for cell growth and invasion. They confirmed their results in mice, and found effects of apigenin on human breast carcinoma may be related to suppression of IL-6 expression. The study is interesting and provided large amount of data.

Questions:

The authors observed several cytokines were increased in MDA-MB-231 cells, why did they only further studies IL-6? The level of IL-8 is much higher than IL-6.

YES, you are very correct. The level of IL-8 is much higher than IL-6 which we already expected. Because increased expression of IL-8 and/or its receptors in many cancers including breast cancer was reported1,2. Also, the studies of IL-8 on triple-negative breast cancer (TNBC) had been already reported in many papers3,4. High IL-6 level on human triple negative breast carcinoma have also been reported, but, the effect of IL-6 on human triple-negative breast carcinoma is controversial up to now5,6, as well as the association between IL-6 and invasiveness of human breast carcinoma has not been clearly understood yet.

Waugh, David J.J; Wilson, Catherin. The interleukin-8 pathway in cancer, Clinical Cancer Research, 2008, 14(21), 6735

Todorović-Raković, Nataša; Milovanović, Jelena. Interleukin-8 in breast cancer progression, J Interferon Cytokine Res. 2013, 33(10), 563-570

Ariane Freud; Corine Chauveau; Jean-Paul BrouilletAnnick LucasMatthieu LacroixAnne Licznar; Françoise Vignon; Gwendal Lazennec. IL-8 expression and its possible relationship with estrogen-receptor-negative status of breast cancer cells, Oncogene, 2003, 22(2), 256-565

Romaine I. Fernando; Marianne D. Castillo; Mary Litzinger; Duane H. Hamilton; Claudia Palena. IL-8 signaling plays a critical role in the epithelial-mesenchymal transition of human carcinoma cells, Cancer Res. 2011, 71(15), 5296-5306

Narmeen Ahmad, Aula AmmarSarah J. StorrAndrew R. GreenEmad RakhaIan O. Ellis, Stewart G. Martin. IL-6 and IL-10 are associated with good prognosis in early stage invasive breast cancer patients, Cancer Immunol Immunother. 2018, 67(4), 537-549 (positive effect of IL-6)

Jin KPandey NBPopel AS. Simultaneous blockade of IL-6 and CCL5 signaling for synergistic inhibitionof triple-negative breast cancer growth and metastasis, Breast Cancer Res. 2018, 20(1), 54 (negative effect of IL-6)

Fig.5b and Fig.7b. Tumor photographs were not taken at the same time, and only one image in Fig.5b showed scale. In Fig.7b, tumors of the same groups seem from several different photographs. The authors should show all individual scales if tumors were not in the same photograph.

In fact, we took the pictures of all tumors at same time. However, each individual image was crafted for the purpose of publication image. We provide a revised manuscript with all image of Figure 5b and Figure 7b including scale bar. However, if you are uncomfortable, we are will to take out figure 5b and 7b from our manuscript.

Round 2

Reviewer 1 Report

All my comments have been corrected/explained.

Reviewer 2 Report

The authors have addressed my questions. I agree the manuscript for publication.